# Evaluation of Hypertension-Related Knowledge, Medication Adherence, and Associated Factors Among Hypertensive Patients in the Aljouf Region, Saudi Arabia: A Cross-Sectional Study

**DOI:** 10.3390/medicina60111822

**Published:** 2024-11-06

**Authors:** Bashayer Farhan ALruwaili

**Affiliations:** Department of Family and Community Medicine, College of Medicine, Jouf University, Sakaka 72388, Saudi Arabia; bfalrwili@ju.edu.sa; Tel.: +966-551913665

**Keywords:** adherence, diet, knowledge, hypertension, patient education

## Abstract

*Background and Objectives:* Hypertensive patients’ knowledge and adherence to prescribed medicines are critical in managing their condition, as poor adherence may lead to adverse cardiac and cerebrovascular events. The present study assessed hypertension-related knowledge and medication adherence among hypertensive patients attending primary health centers (PHCs) in the Aljouf Province, Saudi Arabia. *Materials and Methods:* Using a cross-sectional design, we conducted this survey on 390 patients. Self-reported hypertension knowledge was assessed using the Hypertension Knowledge Level Scale (HK-LS), and medication adherence was determined using the Hill–Bone Medication Adherence Scale. We used binomial regression analysis (adjusted with other variables) to find the associated factors of medication adherence. *Results:* This study found that nearly half (49.2%) of the participants had inadequate knowledge, and poor medication adherence was noted in 40.8% of the participants. We found a positive correlation between HK-LS and the Hill–Bone Medication Adherence Scale scores (Spearman’s rho = 0.312, *p* = 0.002). Medication adherence was significantly associated with job status (private sector (ref: public sector, adjusted odds ratio [AOR] = 2.02, 95% CI = 1.18–3.62, *p* = 0.005)), living in an urban region (ref: rural, AOR = 3.61, 95% CI = 1.85–5.72, *p* = 0.002), and duration since diagnosis of more than 5 years (ref: ≤1 year, AOR = 3.53, 95% CI = 2.36–4.95, *p* = 0.001). *Conclusions:* The present study findings indicate that there is still a critical gap in managing hypertension at the PHCs in this region, and this may lead to poor health outcomes among the patients and burden the healthcare system. Hence, continuous patient education and targeted counseling are recommended for those with poor medication adherence.

## 1. Introduction

Globally, hypertension remains an important public health problem, and about 1.3 billion adults suffer from this chronic condition [1,2]. Nevertheless, the prevalence rate varies across countries, with a higher rate in low- and middle-income nations [3,4]. This scenario is further worsened by the persistent concepts of the “rule of halves” in hypertension [5]. Additionally, this chronic condition is one of the major modifiable risk factors (preventable) for adverse cardiac and cerebrovascular events in adults [6,7]. Similar to the current global situation, this chronic condition is highly prevalent in the Kingdom of Saudi Arabia (KSA); a recent study by Alshammari SA et al. documented that hypertension is highly prevalent (above 22%) among Saudi adults [8].

The management, diagnosis, and prevention of hypertension are influenced by several factors. A lack of understanding and awareness of hypertension and its implications is the main barrier to diagnosis and adequate treatment [9,10]. The existing literature indicates that patients play vital roles in managing hypertension through a holistic approach, including lifestyle changes, regular follow-up, and compliance with the physician’s recommendations [11,12]. According to the International Society of Hypertension and the National Institute for Health and Care Excellence (NICE) in London, adhering to prescribed anti-hypertensive drugs is essential for achieving appropriate hypertension control, which lowers healthcare expenditures and reduces cardiovascular morbidity and mortality [13,14].

Adherence to prescribed hypertensive medication is facilitated by numerous factors [15]. These factors are generally divided into sociodemographic, patient-related (such as knowledge), and available healthcare facilities. In the context of health literacy, patients’ knowledge plays a central role in hypertension control, including medication adherence [16,17]. Numerous authors have used different methods to explore hypertension-related knowledge among patients. However, culturally suitable scales have yielded valid and reliable results [18,19]. A comparative study conducted by Eshah N F et al. among Jordanian participants showed that good knowledge was observed in the majority of their participants [20]. However, their participants showed inadequate knowledge in some domains, such as diet and lifestyle. Similarly, a study conducted by Abu H et al. in two primary clinics found that patients lacked knowledge about the lifelong condition of hypertension [21]. A recent study in 2024 conducted in the southern region of the KSA found that most of the hypertensive patients had poor knowledge overall in several domains, including medication compliance [22]. Health promotion and disease prevention are emphasized as crucial areas to enhance treatment quality and control costs in the Arab world in hypertension awareness campaigns [8]. According to research conducted in Poland, about 85% of patients correctly identified high blood pressure, but over two-thirds did not know that once diagnosed, hypertension is lifelong, and one-third did not know that hypertension can cause kidney impairment [23].

Healthcare related to hypertension has achieved significant advancement with different strategies of screening programs, the availability of new-generation anti-hypertensive medications, and personalized care [24,25]. Nonetheless, medication adherence remains a critical unresolved issue in hypertension control and improving patients’ health outcomes. As mentioned earlier, researchers have noted several barriers and facilitators. Globally, different tools were used to assess the medication adherence among the patients. These tools are either generic or hypertension-specific [26,27,28].

Regardless of the tools used by different authors, the proportion of adherence among hypertensive patients is low across the different countries, and there is a wide variation. This warrants the necessity of region-specific data related to these aspects, which is crucial. Furthermore, due to the dynamic nature of the influencing factors of adherence, it is highly relevant to assess the medication adherence practice and influencing factors regularly to guide practitioners and policymakers in implementing necessary programs or amending existing programs according to the latest available data. In this context, the present study aimed to assess hypertension-related disease knowledge, medication adherence, and associated influencing factors among patients from the Aljouf region, the KSA. Moreover, we determined the correlation between hypertension-related disease knowledge and medication adherence using appropriate statistical tests.

## 2. Materials and Methods

### 2.1. Study Design and Setting

This cross-sectional survey was conducted in the Aljouf Province, the KSA, from May 2024 to September 2024. This province has four main regions: Sakaka, Dwamat Aljindal, Tabarjil, and Qurrayat. According to KSA’s statistics, about half a million people live in this region. The present study includes adult (18 years and above) primary hypertensive patients of Saudi nationals from all the primary health centers (PHCs) of the two regions (Sakaka and Dwamat Aljindal). The survey excluded those who had known secondary hypertension, pediatric hypertensive patients, hospitalized patients, unwilling (to participate) patients, and known psychiatric patients. Furthermore, the author excluded those who managed their hypertension only through lifestyle modifications.

### 2.2. Sampling Description

The author calculated the sample size using an online calculator that followed the principles of standard sample size protocols (n = z^2^pq/e^2^) [29]. In this formula, n = the minimum required sample size. Considering that half of the patients (50%) have good adherence (p = 0.5, and q = 0.5 [1 − p]), 80% power of the study, 95% confidence interval, and 5% margin of error), the authors concluded that to obtain conclusive evidence, there must be a minimum of 384 hypertensive patients participating in the survey. Moreover, this formula was applied to an infinite population. The 50% expected proportion was taken as a standard method to obtain the highest number of samples. We used the convenience sampling method to recruit the study participants. In this method, the patients visiting the selected PHCs were asked to participate in the survey after the follow-up checkup. To ensure that the participants were distributed over the study period, this survey included only five participants per day in one PHC.

### 2.3. Data Collection Procedures

Data collection started after clearance from the Local Committee for Bioethics, Jouf University (No. 13-09-45, dated 12 May 2024). For data collection, the author coordinated with the concerned family physicians from the selected PHCs. For this study, the data collectors were given standardized training to collect data from hypertensive patients. This study followed all the ethical guidelines according to the Declaration of Helinski. After receiving the patients’ informed consent, we requested that they complete the survey through a Google form on data collectors’ electronic devices. The data collection form included three sections. The first section asked for the hypertensive patients’ background information. The second section included 22 hypertension-related questions (ranging from etiology to complication) using a hypertension knowledge level scale (HK-LS) that assessed the patient’s knowledge [20]. The author obtained permission to use the Arabic tool from the corresponding author of the published manuscript. The participants responded to each item as “correct/wrong/not sure”. The correct answers were given as one mark. For further analysis, the total score was summed and converted into a number out of 100. A higher score in the knowledge section indicated better knowledge.

The final section determined the patients’ medication adherence and refill practices. To determine these aspects, we used the Hill–Bone Medication Adherence Scale (9-item). This tool was created by the National Institute of Health and is freely available [30]. Previous studies that used this tool for assessing medication adherence among hypertensive patients stated that this tool is valid and reliable [30,31]. This tool was originally prepared in English. Using appropriate measures, we translated this tool into Arabic. Regarding the questions posed to the patients involved in this section, we asked them to respond on a four-point Likert scale with the following responses: all the time (1 mark), most of the time (2 marks), sometimes (3 marks), and not at all (4 marks). Like the knowledge section, the total marks were added and then transformed into a number within 100. The higher the score, the better the medication adherence practice. The present study’s data collection tool was pre-tested on 31 hypertensive patients to determine cultural adaptability. The Cronbach’s alpha values for the knowledge and medication adherence sections were 0.83 and 0.89, respectively. The knowledge scores were categorized into low (<median) and high (≥median). Regarding adherence, we categorized the scores into poor (less than 80% of the overall score) and good adherence (80% or more of the overall score) [32,33].

### 2.4. Data Analysis

This study’s data were analyzed with the statistical package of social sciences (SPSS V. 21.0). The descriptive data are shown in frequencies and proportions for categorical variables, and the mean and standard deviation (SD) are used to depict the continuous data. The correlation between HK-LS and Hill–Bone Medication Adherence Scale scores was determined using Spearman’s correlation analysis. The predictors of medication adherence (poor vs. good) were evaluated using binomial regression analysis, which is inherently a multivariate analysis method. In this method, the author attempted to evaluate the predictors after adjusting with the covariables of this study. The statistically significant value (alpha) was set as less than 0.05. Furthermore, binomial regression results are depicted as the adjusted odds ratio (AOR), 95% confidence interval (CI) of AOR, and *p*-values.

## 3. Results

During data collection, we asked 446 hypertensive patients to obtain the minimum required participants (390) for this survey (response rate: 87.4%). The background characteristics of the patients are shown in Table 1. Among 390 hypertensive patients, the majority (48.7%) belonged to the age bracket of 46 to 60 years (mean ± SD = 49.5 ± 11.2), were male (55.9%), worked in the public sector (47.2%), were from the urban side (76.7%), were non-smokers (63.3%), had an income from SAR 5000 to 10,000 (36.4%), and had hypertension diagnosed for 2 to 5 years (42.1%).

Participants’ responses in HK-LS and the overall knowledge score are shown in Table 2. The highest proportion of participants responded correctly to the item related to cardiac complications associated with inadequate treatment (93.8%), followed by the necessity of lifelong anti-hypertensive medicine for most cases (89.7%) and cerebrovascular accident (stroke) risk of untreated hypertension (85.6%). However, the highest number of wrong answers were observed in the meat of choice (red meat—54.6% and white meat—52.3%) and the significance of a regular intake of fruits and vegetables (40.5%). The overall mean ± SD of the HK-LS was 15.56 ± 2.28. Of the studied patients, 49.2% had inadequate knowledge.

Participants’ responses to the Hill–Bone Medication Adherence Scale are presented in Table 3. The highest proportion of “not at all” (desired responses) observed was missed in taking anti-hypertensive medicines while feeling better (40.8%) and sick (40.5%). We observed that a sizable proportion of the participants missed their medications (all the time and most of the time) due to forgetfulness (8.2%) and running out of medication (8.2%). The overall mean ± SD of the HK-LS was 28.74 ± 4.46.

According to the cut-off described in the methods, 40.8% of the participants had inadequately (poor) adhered to the prescribed anti-hypertensive medications.

The Spearman’s correlation test revealed that HK-LS and Hill–Bone Medication Adherence Scale scores were positively correlated (rho = 0.314, *p* = 0.002) (Table 4).

The associated factors for medication adherence that were determined using the binomial logistic regression analysis are depicted in Table 5. After adjusting for other covariables of this study, we found that working in private sectors (AOR = 2.02, 95% CI = 1.18–3.62, *p* = 0.005), employment status (AOR = 1.42, 95% CI = 1.25–2.73, *p* = 0.017) (ref: public sector), urban living (ref: rural, AOR = 3.61, 95% CI = 1.85–5.72, *p* = 0.002), smoking status (ref: smokers, AOR = 0.45, 95% CI = 0.32–0.69, *p* = 0.015), duration since diagnosis of more than 5 years (ref: ≤1 year, AOR = 3.53, 95% CI = 2.36–4.95, *p* = 0.001), and adequate knowledge (ref: inadequate, AOR = 2.93, 95% CI = 1.88–4.09, *p* = 0.003) were significant associated factors (predictors) for medication adherence among hypertensive patients.

## 4. Discussion

Hypertension continues to be one of the most significant public health problems worldwide and in the KSA. In addition to appropriate treatment, it is very important for the patients to have extensive knowledge and high adherence to the prescribed medications [1] so that an effective reduction in morbidity and mortality associated with hypertension can be achieved as a part of sustainable development goal 3 (target 3.4) [34]. In this context, this study aimed to assess the level of disease knowledge in various aspects, medication adherence to the prescribed pills, and factors influencing medication adherence in the Aljouf region, the KSA.

Adequate knowledge of all domains of any chronic disease is essential, as the patients are required to manage these conditions for life [16,35]. The present study demonstrated that a higher proportion of correct answers were observed in domains related to complications (such as cardiac, about 94%, and cerebrovascular (stroke risk), about 86%) that may occur for untreated or inadequately managed hypertensive patients. Similar to our findings, some studies from the KSA and other countries, such as Malaysia, found that a higher level of knowledge of the complications was associated with hypertension [20,36,37]. For instance, Alshammari SA et al. reported that their participants provided more than 80% correct responses for both cerebrovascular and cardiac complications [36]. The possible high awareness of complications among patients could be due to sufficient emphasis given by the family physicians on follow-up visits at the PHCs. However, this study found that there was a lower level of correct responses in the domains related to the importance of lifestyle changes, including diet, such as the best choice of meat and the importance of a regular intake of adequate fruits and vegetables. These findings are critical to policymakers’ decisions to consider implementing counseling sessions for patients regarding diet and other lifestyle changes. It is also worth mentioning that Dash S et al. reported that there are numerous barriers to physicians delivering dietary advice to patients [38]. This further supports the present study’s implications on the need for exclusive counseling sessions regarding diet and other lifestyle changes for patients. In contrast to the present study, some other studies have found different findings in assessing the dietary habits required for hypertension patients [38,39]. The possibility of these differences across the study could be differences in cultural settings, the tools used, and the availability of specific healthcare services at the PHCs.

Regarding medication adherence, we found that 6 to 8% of participants responded that they forget “all the time” about most of the items of the Hill–Bone Medication Adherence Scale. Similarly, less than half of the participants responded in all the domains of the scale as “not at all”, which is the desired response for better adherence to the prescribed medicines. This finding is very critical for the family physicians at the primary health centers to ensure the highest proportion of desired responses in medication adherence domains. This can be achieved not only at the time of medical consultation at the clinics but also through teleconsultation as a part of follow-up and continuous patient education. Studies that used the same scale from different regions found varying findings [32,33,40]. Furthermore, using a cut-off value of 80%, we found that about 41% of the participants had poor medication adherence practices. The present survey’s results are similar to those of some studies and contrast with others. For instance, a study by Stanikzai M H et al. found only a slightly higher proportion of non-adherence among their participants [33]. In contrast, Thirunavukkarasu A et al. [41], Khayyat SM et al. [42] from the KSA, Pan J et al. from China [40], and a critical review from the USA [15] found much higher non-adherence among hypertensive patients. These contrasting results across countries and within the regions of the KSA indicate that one size does not fit all, and tailored, region-specific interventions are required to enhance medication adherence. It is worth noting that similar to non-adherence to hypertension, poor medication adherence for diabetes mellitus patients was also observed by some authors [43,44].

The present study highlighted a positive correlation between HK-LS and medication adherence scores. This positive correlation emphasizes the direct and positive effect of enhancing medication that can contribute to better health outcomes among hypertensive patients and decrease the burden on the existing healthcare system. However, patient education must be continuous. This can be achieved through telehealth services, especially in countries like the KSA, where telehealth services are free for citizens [45,46,47]. Similar to our study, a significant association between higher knowledge and better medication adherence was observed in the studies by Farah RI et al. [48] and Abdisa L et al. [49].

Regarding factors associated with medication adherence, the present study observed that those working in the private sector, those living in urban areas, and those who had hypertension had significantly higher levels of medication adherence than others. The possible explanation for the results could be that private sector workers might have continuous health monitoring, the availability of additional incentives, and more commitment to keeping their health in a better position. However, in contrast to this study, some authors found a non-significant association between adherence and residence status [48,49]. Similar to a recent study in the KSA, we found a positive association between medication adherence and residence status [41]. The higher level of adherence among the participants from the urban side could be due to the availability and accessibility of health information resources. The duration of illness is another important factor to be considered in chronic diseases such as hypertension and diabetes, as they might have significant roles in initiating medications, health outcomes, and medication adherence [41,50]. Interestingly, the present study did not find a significant association with age, gender, and education status. However, existing studies have documented differences in influencing factors; some depicted significant associations, and others did not. These differences further emphasize that medication adherence and associated factors are widely varied, and region-specific policies and guidelines are required.

This study used a validated HK-LS and the Hill–Bone Medication Adherence Scale in a unique sociocultural setting. Nonetheless, like any other research, this study also has some limitations. First, hypertensive patients from a single KSA province were included in this study. Therefore, the present study’s conclusions cannot be extrapolated to patients in other provinces of the KSA. This is particularly because variances in knowledge about hypertension and adherence behaviors could be expected due to sociocultural contexts, education, and regional differences in access to healthcare services. Moreover, differences in healthcare providers’ practices across the different facilities (primary health centers and hospitals) and resources across facilities may contribute to patient compliance with prescribed medications, making it difficult to generalize these results. Next, the research participants were sourced from PHCs belonging to the Ministry of Health. Hence, the adherence practices in other healthcare facilities, such as general hospitals and tertiary care centers, may differ. Additionally, due to reliance on self-administered questionnaires, this study could not analyze adherence levels based on specific types of anti-hypertensive medications. Finally, restrictions related to the study design, such as a lack of the ability to identify a temporal association, self-selection bias, and exaggerated responses, must be kept in mind while interpreting the findings.

## 5. Conclusions

The present study observed that more than 40% of hypertensive patients who attended PHCs in the Aljouf region, the KSA, had poor medication adherence. We also found a positive correlation between patients’ hypertension-related knowledge and medication adherence. Medication adherence was significantly higher among certain categories of patients. The present study’s findings indicate that there is still a critical gap in managing hypertension at the PHCs in this region, and this may lead to poor health outcomes among patients and burden the healthcare system. Hence, continuous patient education and targeted counseling for those with poor medication adherence are recommended. This can be achieved not only during medical consultations at clinics but also through teleconsultation as a part of follow-up and continuous patient education. Multi-regional studies should be conducted that explore the qualitative components that lead to poor medication adherence to recognize region-specific variations and explore medication adherence patterns based on anti-hypertensive drug types.

## Figures and Tables

**Table 1 medicina-60-01822-t001:** Background characteristics of the patients (n = 390).

Variables	Frequency	Proportion
Age (mean ± SD)	49.5 ± 11.2	
Age group		
≤45 years	116	29.7
46 to 60 years	190	48.7
>60 years	84	21.5
Sex		
Male	218	55.9
Female	172	44.1
Education level		
Up to high school	175	44.9
Graduate and above	215	55.1
Job status		
Public sector	184	47.2
Private sector	124	31.8
Unemployed	33	8.5
Retired	49	12.6
Residence		
Urban	299	76.7
Rural	91	23.3
Monthly income *		
<5000	116	29.7
5000 to 10,000	142	36.4
>10,000	132	33.8
Smoking status		
Yes	143	36.7
No	247	63.3
Presence of other chronic diseases		
Yes	139	35.6
No	251	64.4
Duration since diagnosis (years)		
≤1	68	17.4
2 to 5	164	42.1
>5	158	40.5
Number of medications		
≤2	175	44.9
>2	215	55.1

* Data mentioned here are shown in Saudi Riyals (SAR) (USD 1 = SAR 3.75).

**Table 2 medicina-60-01822-t002:** Participants’ responses according to the hypertension knowledge level scale (HK-LS) (n = 390).

Items	Correct Response n (%)	Wrong Response n (%)
An elevated diastolic pressure value denotes high blood pressure	293 (75.1)	97 (24.9)
Either systolic or diastolic pressure implies high blood pressure	309 (79.2)	81 (20.8)
Anti-hypertensive pills should be taken every day	317 (81.3)	73 (18.7)
High blood pressure medicines should be taken only when patients feel sick	235 (60.3)	155 (39.7)
In many cases, anti-hypertensive medicines should be taken for life	350 (89.7)	40 (10.3)
People with high blood pressure should take their medication to feel better	266 (68.2)	124 (31.8)
Lifestyle modifications are not required for patients with anti-hypertensive pills	292 (74.9)	98 (25.1)
Hypertension occurs due to growing old. Therefore, pills are not required	217 (55.6)	173 (44.4)
In many cases, if hypertensive patients adopt healthier lifestyle changes, medical treatment may not be necessary	268 (68.7)	122 (31.3)
Hypertensive patients can eat a salt-rich diet if they take medicines as prescribed	308 (79.0)	82 (21.0)
Smoking is not allowed for hypertensive patients	227 (58.2)	163 (41.8)
Hypertensive patients can drink alcohol regularly	300 (76.9)	90 (23.1)
Hypertensive patients must eat fruits and vegetables regularly	232 (59.5)	158 (40.5)
The best way to cook food is frying for patients with high blood pressure	324 (83.1)	66 (16.9)
The ideal cooking choice is boiling or grilling for patients	223 (57.2)	167 (42.8)
The ideal meat variety for high blood pressure patients is lean meat, such as chicken	186 (47.7)	204 (52.3)
The ideal meat option for patients with high blood pressure is red meat, such as beef	177 (45.4)	213 (54.6)
If we do not treat patients properly, they may die earlier due to complications	292 (74.9)	98 (25.1)
If we do not treat patients properly, patients may experience heart disease, such as heart attacks	366 (93.8)	24 (16.4)
If we do not treat patients properly, hypertensive patients may experience renal problems	326 (83.6)	64 (16.4)
If we do not treat patients properly, hypertensive patients may experience strokes (a cerebrovascular accident)	334 (85.6)	56 (14.4)
If we do not treat patients properly, patients may experience vision complications	227 (58.2)	163 (41.8)
Overall score: mean ± SD, median (IQR)	15.56 ± 2.28, 16 (3)	

**Table 3 medicina-60-01822-t003:** Participant responses to the Hill–Bone Medication Adherence Scale (n = 390).

	All the Time n (%)	Most of the Time n (%)	Sometime n (%)	Not at All n (%)
Forgot to take anti-hypertensive tablets	16 (4.1)	16 (4.1)	289 (74.1)	69 (17.7)
Chose not to take blood pressure medicine	24 (6.2)	8 (2.1)	264 (67.7)	94 (24.1)
Forgot to refill anti-hypertensive medications	8 (2.1)	17 (4.4)	289 (74.1)	76 (19.5)
Ran out of blood pressure tablets	16 (4.1)	16 (4.1)	265 (67.9)	93 (23.8)
Skipped blood pressure medication before a doctor’s visit	8 (2.1)	16 (4.1)	256 (65.6)	110 (28.2)
Missed taking blood pressure medicine while feeling better	8 (2.1)	8 (2.1)	215 (55.1)	159 (40.8)
Missed taking blood pressure medicine while feeling sick	16 (4.1)	8 (2.1)	208 (53.3)	158 (40.5)
Missed taking blood pressure medicine due to carelessness	24 (6.2)	8 (2.1)	231 (59.2)	127 (32.6)
Overall score: mean ± SD, median (IQR)	28.74 ± 4.66, 28 (4)			

**Table 4 medicina-60-01822-t004:** Spearman analysis results of HK-LS and Hill–Bone Medication Adherence Scale scores.

Correlation Variables	Spearman’s Rho	*p*-Value (Two-Tailed)
HK-LS vs. Hill–Bone Medication Adherence Scale	0.314	0.002

**Table 5 medicina-60-01822-t005:** Factors associated with medication adherence: test applied—binomial logistic regression (multivariate regression analysis) (n = 390).

Variables	Overall	Good (n = 231)	Poor (n = 159)	Binomial Regression Analysis Findings	
				Adjusted Odds Ratio ** (95% Confidence Interval)	*p*-Value
Age group					
≤45 years	116	57	59	Ref	
46 to 60 years	190	107	83	1.09 (0.48–2.18)	0.149
>60 years	84	67	17	2.31 (0.91–3. 25)	0.074
Sex					
Male	218	117	101	Ref	
Female	172	114	58	0.82 (0.43–1.69)	0.595
Education level					
Up to high school	175	112	63	Ref	
Graduate and above	215	119	96	0.47 (0.23–1.27)	0.091
Job status					
Public sector	184	104	80	Ref	
Private sector	124	83	41	2.02 (1.18–3.62)	0.005
Unemployed	33	17	16	1.41 (1.25–2.73)	0.017
Retired	49	27	22	0.87 (0.63–1.57)	0.203
Residence					
Rural	91	54	37	Ref	
Urban	299	177	122	3.61 (1.85–5.72)	0.002
Monthly income *					
<5000	116	74	42	Ref	
5000 to 10,000	142	83	59	1.78 (0.76–3.12)	0.181
>10,000	132	74	58	2.13 (0.85–4.32)	0.103
Smoking status					
Yes	143	68	75	Ref	
No	247	163	84	0.45 (0.32–0.69)	0.015
Presence of other chronic diseases					
Yes	139	106	33	Ref	
No	251	125	126	1.35 (0.66–2.76)	0.412
Duration since diagnosis (years)					
≤1	68	33	35	Ref	
2 to 5	164	89	75	1.77 (0.83–3.91)	0.537
>5	158	109	49	3.53 (2.36–4.95)	0.001
Knowledge category					
Inadequate	192	98	94	Ref	
Adequate	198	133	65	2.93 (1.88–4.09)	0.003
Number of medications					
≤2	175	110	65	Ref	
>2	215	121	94	1.41 (0.87–3.30)	0.217

* USD 1 = SAR 3.75. ** Adjusted variables in the multivariate analysis: age group, sex, education level, job status, residence, monthly income, smoking status, presence of another chronic disease, duration since hypertension diagnosis, knowledge category, and number of anti-hypertensive medications.

## Data Availability

The raw data supporting the conclusions of this article will be made available by the authors upon request.

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
