# Peer review of "Evaluation of Hypertension-Related Knowledge, Medication Adherence, and Associated Factors Among Hypertensive Patients in the Aljouf Region, Saudi Arabia: A Cross-Sectional Study"

_medicina, 2024, doi:10.3390/medicina60111822_

Round 1

Reviewer 1 Report

Comments and Suggestions for Authors

The article titled “Evaluation of Hypertension-Related Knowledge, Medication Adherence, and Associated Factors among Hypertensive Patients of Aljouf Region, Saudi Arabia: A Cross-Sectional Study” investigates key factors influencing hypertension management in a specific Saudi population. It uses validated tools to measure patient knowledge and medication adherence, providing essential public health data. However, to make a significant contribution to the field, the paper requires several major revisions.

Major Revisions Required:

Lack of Novelty: While the study addresses an important topic, it does not introduce enough new insights into hypertension knowledge or medication adherence. Similar studies have been conducted globally and in Saudi Arabia.

Recommendation: To address the lack of novelty, it is essential to conduct multivariate regression analysis. This would allow for identifying predictors of poor medication adherence and hypertension knowledge levels. By focusing on the factors influencing these outcomes, such as socio-demographic characteristics or clinical factors, the study could offer a more in-depth analysis and contribute valuable new insights.

Inclusion of Medication Details: The current results section lacks specifics on the types of medications prescribed to different patient groups, which is important for understanding adherence patterns.

Recommendation: The authors should precisely list the types of antihypertensive medications prescribed to each group in the results and discuss potential differences in adherence based on the medication type. This would enhance the readers’ understanding of how different treatments affect adherence.

Sub-stratification Based on Medication Type: The study does not explore whether adherence patterns differ based on the type of medication prescribed, which could be an important factor in understanding poor adherence.

Recommendation: Perform an exploratory sub-stratification of adherence levels based on the specific antihypertensive medications prescribed. This secondary analysis could help determine if certain drugs have lower adherence rates, potentially due to adverse events or other reasons. Understanding these patterns could offer actionable insights for improving medication adherence.

Graphical Abstract: The study lacks visual summarization of its key findings, which can help readers quickly grasp the most important points.

Recommendation: Construct a graphical abstract to visually represent the relationship between hypertension knowledge, medication adherence, and their associated factors. This could include visual data on adherence rates, significant predictors, and potential interventions, making the paper more engaging and accessible.

Methodological Precision: The binomial regression analysis used needs further justification and clarity regarding its selection.

Recommendation: Clarify why binomial regression was chosen and specify why particular variables were adjusted for. Additionally, using a multivariate regression model will provide a more comprehensive analysis by accounting for multiple confounding factors, offering a more robust examination of predictors.

Discussion Expansion: The discussion of the results is somewhat limited and needs to be broadened to consider how different antihypertensive medications might influence adherence.

Recommendation: Expand the discussion to explore the possible reasons behind varying adherence levels, such as the impact of side effects or complexity of medication regimens. The authors should also suggest how the findings could guide clinical practice and health policy to improve adherence, particularly in the Saudi context.

Incorporating these revisions—especially the multivariate regression analysis, medication-specific sub-stratification, and a graphical abstract—will significantly enhance the paper’s rigor and novelty, making it a more valuable contribution to the field.

Comments on the Quality of English Language

Please check the quality of the language with the native speaker.

Reviewer 2 Report

Comments and Suggestions for Authors

The article entitled "Evaluation of Hypertension-Related Knowledge, Medication Adherence, and Associated Factors among Hypertensive Patients of Aljouf Region, Saudi Arabia: A Cross-Sectional Study" examines the knowledge of hypertension and medication adherence of patients attending primary health centres (PHCs) in Aljouf Province, Saudi Arabia. The study's compliance with the Declaration of Helsinki is remarkable and commendable. However, there are some methodological and reporting issues that need to be addressed to improve the clarity and robustness of the results.

The study sample is from a single geographical region (public health centres in Aljouf), which raises concerns about the generalizability of the results. Hypertensive patients from different regions or health facilities may have different levels of knowledge and adherence to treatment, which limits the generalizability of the results to wider populations. Further studies with a more diverse sample would strengthen the external validity of the conclusions.

The authors mention the use of an online calculator to determine the sample size, but fail to provide important details such as the total number of participants included, the margin of error. Providing this information is essential for assessing the adequacy of the sample size and the statistical power of the study.

Certain abbreviations, such as PHC and LCBE, are introduced without explanation the first time they appear in the text (except in the summary). It is important that you spell out each abbreviation the first time it appears to ensure clarity for all readers.

There are notable discrepancies between the data presented in the tables and the figures given in the text. For example:

Line 165: the authors state that the percentage of non-smokers in the sample is 36.7%, whereas the correct figure according to the data appears to be 63.3%.

Line 177: The percentage of incorrect answers regarding the "meat of choice" is given as 52.3%, although the data suggest that it should be 54.6%.

Line 228 (Discussion): The authors claim that the study found a higher proportion of correct answers in relation to complications such as heart problems (94%) and cerebrovascular risks (86%). However, the results presented do not clearly support this conclusion, so it is difficult to substantiate this statement based on the available data.

While the findings on inadequate knowledge and poor medication adherence in hypertensive patients point to an important public health issue, the discussion could benefit from a more nuanced interpretation of the data. In particular, the discrepancies between the reported percentages and the actual results need to be reconciled to ensure that the conclusions drawn are based on the evidence presented.

Round 2

Reviewer 1 Report

Comments and Suggestions for Authors

Dear Authors,

The quality is a little bit improved; however, not sure if adequate for Medicina journal.

I am going to suggest the Editors to ask for additional peer-reviewer on top of my opinion.

Thank you for the modifications made (other not made would make the difference regarding novelty and quality of evidence).

Best regards.

Reviewer 2 Report

Comments and Suggestions for Authors

The authors have improved the manuscript and addressed the issues that were raised. Regarding the question concerning the variability in knowledge and adherence levels to treatment among hypertensive patients from different regions or healthcare facilities, which limits the generalizability of the findings to broader populations, I would suggest further highlighting this limitation in the discussion. While the authors do mention this point, the reference is rather brief; in my opinion, this limitation should be more emphasized to strengthen the study’s contextual clarity

Comments on the Quality of English Language

The English could be improved to more clearly express the research.
